# Magnetic Nanoparticle Support with an Ultra-Thin Chitosan Layer Preserves the Catalytic Activity of the Immobilized Glucose Oxidase

**DOI:** 10.3390/nano14080700

**Published:** 2024-04-17

**Authors:** Boris B. Tikhonov, Daniil R. Lisichkin, Alexandrina M. Sulman, Alexander I. Sidorov, Alexey V. Bykov, Yury V. Lugovoy, Alexey Y. Karpenkov, Lyudmila M. Bronstein, Valentina G. Matveeva

**Affiliations:** 1Department of Biotechnology, Chemistry and Standardization, Tver State Technical University, 22 A. Nikitina Str., 170026 Tver, Russia; tiboris@yandex.ru (B.B.T.); danok9900@gmail.com (D.R.L.); alexsulman@mail.ru (A.M.S.); sidorov_science@mail.ru (A.I.S.); bykovav@yandex.ru (A.V.B.); pn-just@yandex.ru (Y.V.L.); 2Department of Condensed Matter Physics, Tver State University, Zhelyabova St., 33, 170100 Tver, Russia; karpenkov_alex@mail.ru; 3Department of Chemistry, Indiana University, 800 E. Kirkwood Av., Bloomington, IN 47405, USA

**Keywords:** biocatalyst, magnetite, nanoparticle, chitosan, glucose oxidase, magnetic nanoparticle aggregates

## Abstract

Here, we developed magnetically recoverable biocatalysts based on magnetite nanoparticles coated with an ultra-thin layer (about 0.9 nm) of chitosan (CS) ionically cross-linked by sodium tripolyphosphate (TPP). Excessive CS amounts were removed by multiple washings combined with magnetic separation. Glucose oxidase (GOx) was attached to the magnetic support via the interaction with N-hydroxysuccinimide (NHS) in the presence of carbodiimide (EDC) leading to a covalent amide bond. These steps result in the formation of the biocatalyst for D-glucose oxidation to D-gluconic acid to be used in the preparation of pharmaceuticals due to the benign character of the biocatalyst components. To choose the catalyst with the best catalytic performance, the amounts of CS, TPP, NHS, EDC, and GOx were varied. The optimal biocatalyst allowed for 100% relative catalytic activity. The immobilization of GOx and the magnetic character of the support prevents GOx and biocatalyst loss and allows for repeated use.

## 1. Introduction

Magnetically recoverable biocatalysts, i.e., enzymes immobilized on magnetic supports, receive considerable attention due to the important advantages of such catalysts compared to native enzymes or enzymes immobilized on nonmagnetic supports (see recent reviews [1,2,3,4,5,6,7,8,9,10,11,12,13,14]). Like all immobilized enzymes, they allow for higher stability during catalytic reactions and broader temperature and pH stability ranges [15,16,17,18,19]. In addition, magnetic supports grant easy magnetic separation during both the catalyst synthesis and the catalytic reaction as well as magnetic control for continued processes [20,21,22,23,24,25]. It is noteworthy that magnetic nanoparticle (NP) morphology is crucial for magnetic properties. For example, cubic [26] or rod-shape [27] NPs could exhibit different saturation magnetization than that of spherical NPs of the same size [28,29,30]. In several studies, cubic NPs showed the highest saturation magnetization [29,30]. However, when magnetic NP aggregates are formed, the shape influence is secondary compared to the combined properties of an NP ensemble.

When magnetic supports are based on magnetic NPs, the latter require functionalization for the further covalent attachment of enzymes. A typical approach is coating magnetic NPs with silica functionalized with amino groups [31,32,33,34]. Amino groups allow for the chemical attachment of enzymes via suitable linkers. Unlike silica-NH_2_ treatment, coating magnetic NPs with functional polymers (i.e., polymers containing functional groups) can provide a much higher functional group density in one step via polymer deposition on the NP surface. Among functional polymers, significant attention has been paid to chitosan (CS)—a natural polymer obtained from chitin (biomass) [8,35,36,37,38,39]. It was reported that the CS coating of maghemite NPs can be carried out by different methods. The authors of ref. [38] explored a microemulsion process, a suspension cross-linking process, and a covalent binding method. They discovered that the suspension cross-linking process provides the most efficient biocatalyst, which was assigned to the preferable orientation of amino groups for enzyme immobilization. Gracida et al. reported the deposition of CS on magnetite NPs by the simple dispersion of both components in an ultrasound processor which demonstrates a simpler method than those described above, yet it allows for an efficient biocatalyst after cross-linking with genipin and the immobilization of xylanase [36]. An even more robust method was reported by Salehi et al. when magnetite NPs were dispersed in the CS solution and separated from the suspension after polymer adsorption [39]. We used a similar method in our work reported here.

It is worth noting that the stabilization of CS on the support can be improved via CS cross-linking. As cross-linking agents, a number of compounds have been utilized including glutaraldehyde [40,41,42], genipin [36,43,44], and sodium tripolyphosphate (TPP) [45,46,47]. Glutaraldehyde is a popular cross-linking agent because its aldehyde groups immediately form covalent bonds with amino groups. The use of this cross-linker is often warranted if the material application is not sensitive to glutaraldehyde shortcomings [40]. In genipin, an aldehyde group is formed upon interaction with an amino group via the opening of the dihydropyran ring followed by the reaction of an aldehyde with a secondary amino group, resulting in covalent bonding [36,43,44]. Genipin is often preferred in biomedical applications when its high cost is not an issue [43]. Chandra et al. demonstrated the formation of ionic cross-linking between CS-protonated amino groups (−NH_3_^+^) and tri-polyphosphate anionic groups of TPP on the surface of magnetite NPs for the removal of heavy metals from wastewater [45]. Chitosan-TPP composites have been also used as nanofertilizers to deliver nutrients to plants [46]. Considering that glutaraldehyde is toxic and cannot be used for pharmaceutical applications [48] while genipin is too expensive, TPP is a compound of choice for CS cross-linking. It provides ionic interactions with protonated CS amino groups in mild conditions.

The other avenue to stabilize a CS layer was demonstrated for porous supports. An uneven surface (porosity) improves CS adsorption, thus stabilizing the composite [49]. Moreover, enzyme adsorption was also governed by the pore sizes and total porosity. In general, enhanced porosity is beneficial for biocatalysts due to a possible higher degree of enzyme attachment [50,51].

Among numerous enzymes, glucose oxidase (GOx) belonging to the oxidoreductase family plays an important role in biological processes. In the presence of oxygen, GOx catalyzes the oxidation of D-glucose to D-glucono-δ-lactone and hydrogen peroxide. In turn, D-glucono-δ-lactone is further spontaneously hydrolyzed to D-gluconic acid [52,53,54]. D-gluconic acid and its derivatives are widely utilized in the food, pharmaceutical, textile, and construction industries [55,56].

In this paper, we developed magnetically responsive biocatalysts for the oxidation of D-glucose to D-gluconic acid by first forming magnetite (Fe_3_O_4_) NP aggregates via a precipitation route at ambient temperature [57,58]. This method uses the least expensive chemicals and allows for the conservation of energy compared to the thermal decomposition methods of metal compounds frequently used for syntheses of magnetic NPs [59,60,61]. Here, the magnetic NP aggregates (MNAs) were then coated with an ultrathin layer of CS cross-linked with a most benign compound—TPP—which is allowed for use in the food and pharmaceuticals industries [62]. In addition, the attachment of GOx was carried out with benign 1-ethyl-3-(3-dimethylaminopropyl)carbodiimide hydrochloride (EDC) and N-hydroxysuccinimide (NHS) [63,64]. The optimization of the biocatalyst composition allowed us to achieve 100% relative activity, paving the way for commercial applications of such catalysts.

## 2. Materials and Methods

Materials, synthesis of MNAs, biocatalyst activity assays, reusability of biocatalysts, and characterization are presented in the Appendix A.

### 2.1. Magnetic Biocatalyst Synthesis

MNAs were synthesized following the procedure described elsewhere by forming magnetite nanoparticle aggregates by co-precipitation of FeCl_2_ (ferrous chloride) and FeCl_3_ (ferric chloride) in basic medium [57]. A typical experiment is described in the Appendix A.

To cover the MNAs with CS, in a typical experiment, 10 mL of CS solution (0.1 g in 10 mL of 2 M acetic acid) was added to the MNA reaction mixture (without separation of MNAs) and stirred for 15 min. After that, MNA-CS was separated from the reaction mixture using a rare earth magnet. For CS cross-linking on the MNA surface, the MNA-CS was stirred for 1 h in a solution of TPP (0.05 g in 50 mL of distilled water), after which the MNA-CS was washed five times, separated with a rare-earth magnet, and dried in air at 20 °C for 24 h. The resulting sample was designated as MNA-CSP where CS stands for chitosan and P stands for TPP.

For the covalent attachment of GOx with the support, in this work we used EDC and NHS [65,66,67]. When these reagents are added and kept in the reaction mixture for 12 h, a stable NHS ester is formed on the surface of GOx. This resulted in the formation of a durable amide bond after interaction with the NH_2_ groups on the support surface.

In a typical experiment, the dry MNA-CSP sample (0.25 g) was added to 0.1 g of EDC, 0.04 g of NHS, and 50 mg of GOx in 20 mL of PBS buffer (pH 6) and stirred for 12 h. Then, the biocatalyst (MNA-CSP-GOx) was magnetically separated, washed five times with 50 mL of water each, and dried at 20 °C for 24 h.

### 2.2. The Catalytic Reaction

The catalytic reaction of choice for immobilized GOx is D-glucose oxidation to D-gluconic acid with peroxidase (HRP) and 2,2′-Azino-bis(3-ethylbenzothiazoline-6-sulfonic acid) diammonium salt (ABTS) to carry out analyses by spectrophotometry (Appendix A) [68]. When O_2_ is present, the biocatalyst based on GOx allows for oxidation of D-glucose first to D-glucono-1,5-lactone and H_2_O_2_. After that, the lactone is hydrolyzed to D-gluconic acid (Appendix A). Because D-glucono-1,5-lactone and H_2_O_2_ are formed in equimolar amounts (also equal to the D-glucose moles), we assessed the amount of H_2_O_2_ at specific times with spectrophotometry (a colored compound is formed in the presence of ABTS and H_2_O_2_ (Appendix A) [69].

## 3. Results and Discussion

### 3.1. Characterization of MNAs

Figure 1 displays bright-field (BF) scanning transmission electron microscopy (STEM) and high-resolution TEM (HRTEM) images of MNAs. The STEM image shows a large aggregate consisting of smaller NPs attached to each other (Figure 1a). The formation of NP aggregates is typical for the precipitation method which is carried out in the absence of efficient stabilizing molecules such as surfactants or polymers [57,58]. We estimated mean particle sizes using the grain intercept method. For this, we used about 30 NPs of each sample. The aggregates are comparatively large (~up to several microns) and polydisperse. The individual NP sizes vary between 5 nm and 30 nm with a mean diameter of about 12 nm. However, because all NPs are glued to each other, these measurements are not accurate.

The HRTEM image of MNAs (the [111] orientation) displays the interplanar distances of d(220)_Fe_3_O_4__ = 0.29 nm (Figure 1b) which are typical for the [111] projection of the magnetite (Fe_3_O_4_) crystal structure [57]. Hysteresis loops obtained at two different temperatures (Appendix A) show that the saturation magnetization of MNAs is 65.3 emu/g which provides a high magnetic response—a clear advantage of MNAs vs. individual magnetic NPs.

### 3.2. Biocatalyst Catalytic Activity

The biocatalyst synthesized (MNA-CSP-GOx) is a multicomponent system, containing MNAs, CS ionically cross-linked by TPP, and GOx covalently attached to CS via interaction with NHS in the presence of EDC. To discover the optimal contents of all components producing the most efficient catalyst, we varied the amounts of CS, TPP, cross-linkers for GOx (NHS and EDC), and GOx. The data obtained are presented in Table 1.

The CS amount was varied between 0.05 g and 0.125 g per 1 g of MNAs (Table 1). The maximum relative activity (R), i.e., activity towards native GOx in the same conditions, was obtained for 0.1 g of CS. A decrease in the CS amount leads to a decrease in the relative activity which could be assigned to an insufficient number of CS amino groups. At the same time, an increase in the CS amount leads to an even faster drop in activity which could be tentatively assigned to a too thick CS layer which could bury GOx, changing its native conformation.

TPP, which is introduced after CS deposition, serves as cross-linker for CS stabilization (Figure 1). The TPP amount varied between 0.01 g and 0.1 g per 0.1 g of CS. It was found that the best activity was observed for 0.05 g of TPP, which could be attributed to a combination of sufficient stability of the CS layer during reaction and adequate access to CS amino groups.

For immobilization of GOx, in this work we used a premade mixture of EDC and NHS as a cross-linker instead of glutaraldehyde utilized by us earlier [70]. Glutaraldehyde is too sensitive and too reactive for successful reaction control in addition to its toxicity. The EDC and NHS mixture leads to the formation of the NHS ester on GOx which, in turn, forms a stable amide group with the CS NH_2_ groups (Figure 1).

The amounts of NHS and EDC varied from 0.02 g of EDC + 0.008 g of NHS to 0.125 g of EDC + 0.05 g of NHS (Table 1). The optimal amount was 0.1 g of EDC + 0.04 g of NHS.

Thus, we identified the composition of the optimal catalyst, MNA-CSP-GOx-3, providing 100% relative activity in D-glucose oxidation. It is noteworthy that 100% relative activity of an immobilized enzyme has been already reported by a few authors including us [68,70,71], but in no case was it accomplished with a magnetically recoverable benign biocatalyst suitable for the food and pharmaceutical industries.

### 3.3. Characterization of MNA-CSP-3

The structure and properties of the optimal catalyst were investigated using a combination of physicochemical methods. As was expected, coating with CS and cross-linking with TPP does not change the iron oxide morphology and phase composition (Figure 2). The HRTEM image corresponds to [001] projection of magnetite with (220) crystal planes.

To assess the presence of CS, the MNA-CSP-3 sample was evaluated by energy dispersive spectroscopy (EDS) (Figure 3), X-ray photoelectron spectroscopy (XPS) (Appendix A), and thermal gravimetric analysis (TGA) (Figure 4).

EDS and XPS spectra clearly demonstrate the presence of C and P, indicating the cross-linked CS layer. Nitrogen is only detected by XPS, which is not surprising. XPS is a surface method, while EDS is a volume method, so the latter is less sensitive to a small amount of an element (nitrogen) on the surface of MNA-CSP. The TGA data (Figure 4) of MNAs, MNA-CSP-3, and pure CS allowed us to calculate the thickness of the CS layer. Taking into account the weight loss of 3.6% (due to volatiles) from initial MNAs, 9.3% of the weight loss from MNA-CSP-3, and the fact that pure CS loses only 63% of its weight at 600 °C, we calculated that the sample contains 8.9% of CS, i.e., 0.089 g of CSP per 1 g of MNA-CSP. Considering that chitosan density is 1g/cm^3^ [72] and the MNA surface area (see data below) is 98 m^2^/g, the CS layer thickness is 0.9 nm. To the best of our knowledge, this is one of a few examples of a well-documented ultra-thin CS layer deposited in such a simple procedure [38,73,74].

Hysteresis loops presented in Appendix A show that the saturation magnetization of MNA-CSP-3 decreases only by 15% (to 55.4 emu/g) after the deposition of the cross-linked CS layer, thus allowing for easy magnetic separation both during the biocatalyst synthesis and after the catalytic reaction.

An important feature of the MNA support is its porosity with a surface area of 98 m^2^/g and pore volume of 0.32 cm^3^/g (Appendix A, Appendix A). The adsorption–desorption isotherms of MNAs, MNA-CSP, and MNA-CSP-GOx (Appendix A) belong to type IV, which is typical for mesoporous materials [75]. The major porosity is provided by mesopores with an average size of 16 nm (Appendix A) most likely due to cavities formed within MNAs by interconnecting magnetite NPs. Despite the ultra-thin CS layer (determined by the TGA data) in the MNA-CSP sample, the surface area and pores sizes are somewhat lower than those in MNAs, which could be assigned to the deposition of CS in some pore junctions.

### 3.4. Analysis of GOx Activity

We explored two kinetic parameters—the Michaelis constant (*K_m_*) and the maximum reaction rate (*V_max_*)—to assess the activity of immobilized GOx. In particular, *K_m_* describes the interaction between the enzyme and substrate, while the reaction rate is described by *V_max_* [69].

We performed kinetic tests to assess the above parameters of free GOx and MNA-CSP-GOx-3 (see Appendix A and the adjacent text in the Appendix A). Then, we plotted the dependence of the reaction rates (*V* μM/s) on the concentration of D-glucose (*S*, mM) followed by the Lineweaver–Burk graphs (Appendix A) to calculate kinetic parameters (*K_m_*, *V_max_*) [76].

The data presented in Table 2 indicate that for MNC-CSP-GOx-3, *V_max_* is lower than that for native GOx. This could be caused by diffusion at the interface between solid and liquid because the immobilized enzyme is a heterogenous catalyst. At the same time, *K_m_* for the biocatalyst is also lower than that for native GOx. This is not always the case as is illustrated in ref. [77]. The lower *K_m_* validates better affinity between GOx and the substrate and higher GOx conformational stability when the reaction occurs with GOx immobilized on the ultra-thin CS layer.

It is well known that enzyme activity strongly depends on pH. Active centers of GOx contain ionogenic groups, whose conformation and condition can change depending on pH [78,79]. Figure 5a, showing the dependence of the relative activity of free and immobilized GOx on pH (in the 3–12 pH range), demonstrates that for MNA-CSP-GOx-3, the activity is approximately 10% higher in a wide pH range (standard deviations are 2–3%) than that for native GOx. This could be attributed to the better conformational stability of GOx in MNA-CSP-GOx-3 upon pH change (the optimal pH is 6).

To evaluate the influence of the temperature on the relative activities of MNA-CSP-GOx-3, we used the 20–80 °C temperature range at pH 6 for correlation with native GOx (Figure 5b). For native GOx, the maximum activity is achieved at 35 °C, but then it drops at higher temperatures, most likely due to denaturation. MNA-CSP-GOx-3 displays better tolerance to temperature variations. The GOx stabilization observed in this work is similar to that in the data obtained by other authors [80,81].

### 3.5. Repeated Use and Stability of the Biocatalyst

The important advantage of magnetically recoverable biocatalysts is easy reuse without catalyst loss, which opens opportunities for commercial applications. But such repeated use of the biocatalysts is only justified if their catalytic performance is stable [68].

We carried out ten successive experiments with the same biocatalyst load in the D-glucose oxidation at pH 6 and 35 °C (optimal conditions). At the end of each reaction, the biocatalyst was magnetically separated for about 30 sec and then employed again. The sequence was repeated three times for statistical analysis. The data (Figure 6, Appendix A) show that the relative activity of MNA-CSP-GOx-3 decreases by only 9% after five successive experiments and by 20% after ten.

In addition to reuse, we carried out long-term incubation stability experiments of MNA-CSP-GOx-3 in comparison with native GOx. For this, the relative activity of GOx and MNA-CSP-GOx-3 were determined at certain time intervals for 80 h at 25 °C and 90 days at 4 °C. Figure 7a shows that free GOx maintained only 25.6% of its activity after 40 h, while the relative activity of MNA-CSP-GOx-3 was 79%. After 80 h, the activity of native GOx dropped to 5.3%, while MNA-CSP-GOx-3 retained 60.3% of its activity. It was demonstrated that the decrease in the enzyme activity at room temperature can be assigned to the protein decay due to bacterial growth [82,83]. For the 4 °C measurements (bacterial growth is avoided (Figure 7b)), native GOx activity remains high for five days and then drops to 55.1% after 90 days (probably due to denaturing). MNA-CSP-GOx-3 displayed a decrease to only 78.9% after 90 days—an excellent stability. We think that the exceptional storage stability of MNA-CSP-GOx-3 could be assigned to the preservation of optimal GOx conformation in the CS layer due to limited denaturing.

## 4. Conclusions

In this work, we developed magnetically responsive biocatalysts by the formation of an ultra-thin cross-linked CS layer on the surface of MNAs. The immobilization of GOx was carried out using benign NHS (a linker) in the presence of EDC. We believe that the important avenue here is the optimization of the bioca0talyst composition by varying the amounts of all biocatalyst components and testing the catalytic properties of the biocatalysts synthesized in the oxidation of D-glucose to D-gluconic acid. The best biocatalyst, MNA-CSP-GOx-3, was prepared with 0.1 g of CS, 0.05 g of TPP, 0.1 g of NHS, 0.04 g of EDC, and 0.01 g of GOx per 1 g of MNAs. This biocatalyst provided 100% relative activity at pH 6 and 35 °C in D-glucose oxidation. The data obtained allow us to suggest that an ultra-thin CS layer and a suitable linker for GOx provide the preservation of the enzyme conformation and the functioning of the immobilized GOx as native GOx. The kinetic tests demonstrate better affinity between GOx and the substrate and higher GOx stability when the reaction occurs with GOx immobilized on the ultra-thin CS layer than for native GOx. Magnetic support (MNA) allows for easy magnetic separation and successful reuse. The simplicity of the MNA formation and benign constituents utilized in the fabrication of this biocatalyst make it suitable for the pharmaceutical industry. We believe that the proposed approach is universal and can be utilized for the immobilization of other enzymes to fabricate biocatalysts for many important enzymatic processes.

## Data Availability

Research data are available from authors upon request.

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
