# Peer review of "Magnetic Nanoparticle Support with an Ultra-Thin Chitosan Layer Preserves the Catalytic Activity of the Immobilized Glucose Oxidase"

_nanomaterials, 2024, doi:10.3390/nano14080700_

Round 1

Reviewer 1 Report

Comments and Suggestions for Authors

The manuscript Nanomaterials - 2961466 focuses on the novel hybrid magnetic nanoparticles and their application as carriers for a glucose oxidase. The carriers and immobilised biocatalysts are thoroughly characterised.

The topic is interesting and of current scientific and practical interest.

The study is well motivated. The experimental design and the chosen methodology for the characterisation of the nanoparticles are appropriate. The experimental part is described in sufficient detail. The results are clearly presented in six figures and two tables and are critically discussed and analysed.

In my opinion, the introduction should be expanded. It contains 59 references, which are only mentioned without giving a specific background or setting the scientific problem.

The conclusion is indeed a summary of the research. It should be revised. 

Author Response

Comment 1: In my opinion, the introduction should be expanded. It contains 59 references, which are only mentioned without giving a specific background or setting the scientific problem.

Response 1: The Introduction has been expanded.

Comment 2: The conclusion is indeed a summary of the research. It should be revised. 

Response 2: The conclusion has been rewritten.

Reviewer 2 Report

Comments and Suggestions for Authors

In their manuscript entitled “Magnetic nanoparticle support with ultra-thin chitosan layer preserves catalytic activity of the immobilized glucose oxidase“, the authors describe a facile route for the preparation of an enzymatic catalyst with magnetic nanoparticle aggregates as a support. The composition of this supported catalyst has been optimized and characterized for their composition, structure, surface area, and enzymatic activity. The research was complemented by the proof of the recyclability of the supported catalyst.

I do not find any objection for a publication in nanomaterials after considering the following suggestions by the authors:

In line 176 (page 6) it is mentioned that “both spectra” (i.e., EDS in Figure 3 and XPS in Figure S2) demonstrate the presence of C, N and P. However, the EDS spectrum in Figure 3 does not indicate the presence of nitrogen. Furthermore, it would be beneficial to display high-resolution XPS spectra for the various elements, especially for the peak of nitrogen, which appears diminishingly small in the survey spectrum in Figure S2.

Make sure to combine the lines of Table 2 (pages 7-8) in the final layout of the manuscript.

I find Figures S4 and S5 more relevant than Figure 5 and suggest to exchanges these figures between the main text and the supporting information.

The review of literature is impressive. I did not verify the individual references. For reference 67, the year of access of this website is missing (accessed on March 7, 2024?).

Author Response

Comment 1: In line 176 (page 6) it is mentioned that “both spectra” (i.e., EDS in Figure 3 and XPS in Figure S2) demonstrate the presence of C, N and P. However, the EDS spectrum in Figure 3 does not indicate the presence of nitrogen. Furthermore, it would be beneficial to display high-resolution XPS spectra for the various elements, especially for the peak of nitrogen, which appears diminishingly small in the survey spectrum in Figure S2.

Response 1: We are thankful to the reviewer for noticing an incorrect statement. Indeed, the EDS spectrum does not show nitrogen as it is a volume method, and the nitrogen content is low (EDS spectrum figure was replaced for the better version in the revised manuscript). We corrected the discussion pertinent to this subject (see page 6, lines 208-211). The high-resolution (HR) XPS spectrum of nitrogen shows a weak, broad signal and it is not informative. Other HR XPS spectra do not show any useful information either. That is why they are not discussed.

Comment 2: Make sure to combine the lines of Table 2 (pages 7-8) in the final layout of the manuscript.

Response 2: The final layout is made by the publisher, and I believe it will be corrected.

Comment 3: I find Figures S4 and S5 more relevant than Figure 5 and suggest to exchanges these figures between the main text and the supporting information.

Response 3: Figures have been exchanged as the reviewer suggested.

Comment 4: The review of literature is impressive. I did not verify the individual references. For reference 67, the year of access of this website is missing (accessed on March 7, 2024?).

Response 4: The reference has been corrected.

Reviewer 3 Report

Comments and Suggestions for Authors

In this study, magnetite nanoparticles coated with CS cross-linked by TPP. GOx was attached to magnetic support via the interaction with NHS and EDC. Authors demonstrate the formation of biocatalyst for D-glucose oxidation to D-gluconic acid. Overall, this GOx immobilization could prevent the loss of GOx and biocatalyst loss. Major concerns can be found as follows.

1. The successful conjugation of GOx on MNA-CSP should be characterized by FTIR.

2. GOx activity needs to be assessed in a long-term and under different ambient conditions (pH, temperature, etc.).

3. The catalysis kinetics of GOx before and after immobilization needs to be depicted by KM function curves.

4. The structural conformation of GOx after immobilization onto MNA needs to be further characterized.

Author Response

Comment 1:  The successful conjugation of GOx on MNA-CSP should be characterized by FTIR.

Response 1: We agree with the reviewer that a FTIR spectrum can provide useful information about an organic phase. However, because the chitosan layer is so thin (0.9 nm), the ATR FTIR spectrum obtained by us showed unresolved bands. That is why FTIR was not included in this manuscript.

Comment 2: GOx activity needs to be assessed in a long-term and under different ambient conditions (pH, temperature, etc.).

Response 2:  The data on long-term activity have been added on page 7 (lines 283-295) and on page 8 (Figure 7).

Comment 3:  The catalysis kinetics of GOx before and after immobilization needs to be depicted by KM function curves.

Response 3:  KM function curves have been added in Supplementary Materials.

Comment 4: The structural conformation of GOx after immobilization onto MNA needs to be further characterized.

Response 4:  Normally, GOx conformation is indirectly characterized by fluorescence, circular dichroism, and NMR spectroscopy. When GOx is immobilized on the surface of magnetic NPs, such a characterization is impossible. At the same time, immobilized GOx performance can be considered as an another indirect characterization of conformation. According to literature (Francesco Secundo. Conformational changes of enzymes upon immobilization. Chem. Soc. Rev., 2013, 42, 6250-6261), “the most sensible method to detect changes in the catalytic machinery of an immobilized enzyme is the measurement of its activity”.

Reviewer 4 Report

Comments and Suggestions for Authors

The manuscript titled “Magnetic nanoparticle support with ultra-thin chitosan layer preserves catalytic activity of the immobilized glucose oxidase” by Tikhonov, B.B.; et al. is a scientific work where the authors studied the catalytic performance of magnetite magnetic nanoparticle aggregates coated by chitosan chains. The transformation of substrate (D-glucose) to D-gluconic acid mediated by glucose oxidase (GOx) in presence of horseradish peroxidase enzymes was monitored for this purpose. Many complementary techniques were devoted to fully characterize the properties of the examined grafted magnetite magnetic nanoparticles. The manuscript is generally well-written and this is a topic of growing interest.

However, it exists some points that need to be addressed (please, see them below detailed point-by-point) to improve the scientifc quality of the submitted manuscript paper before this article will be consider for its publication in Nanomaterials.

1) ABSTRACT. “This results in (…)” (line 19). Please, the authors should modify this statement by “These results in (…)”. Even if the manuscript is generally well-written some revision is required in order to polish final details.

2) KEYWORDS. The authors should consider to add the term “magnetic nanoparticle aggregates” in the keyword list.

3) INTRODUCTION. “When magnetic supports are based on magnetic nanoparticles (NPs) (…) amino groups” (lines 36-38). Here, even if I agree with the authors some further discussion needs to be devoted in order to highlight the importance of the shape of magnetic nanoparticles and how other alternative morphologies like cubic [1] or rod-shape [2] exhibit lower magnetic responses [3] than the shape chosen by the authors. This will significantly aid to the potential readers to better understand the significance of the magnetic nanoparticle synthesis strategy chosen by the authors.

[1] Wang, Q.; Ma, X.; Liao, H.; Liang, Z.; Li, F.; Tian, J.; Ling, D. Artificially Engineered Cubic Iron Oxide Nanoparticles as a High-Performance Magnetic Particle Imaging Tracer for Stem Cell Tracking. ACS Nano 2020, 14, 2053-2062. https://doi.org/10.1021/acsnano.9b08660.

[2] Marcuello, C.; Chambel, L.; Rodrigues, M.S.; Ferreira, L.P.; Cruz, M.M. Magnetotactic Bacteria: Magnetism Beyond Magnetosomes. IEEE Trans. Nanobioscience 2018, 17, 555-559. https://doi.org/10.1109/TNB.2018.2878085.

4) “In this paper (…) magnetite (Fe3O4) NP aggregates via a precipitation route (…) magnetic NPs” (lines 60-64). What are the main advantages of magnetic NP aggregates in comparison to indiviual magnetic NP? A brief discussion should be furnished in this regard.

5) MATERIALS AND METHODS. “MNA (…) co-precipitation of FeCl2 and FeCl3 in (…)” (lines 77-79). Please, the authors should state the full-name of all chemical compounds the first time that they appear in the manuscript. This comment should be taken into account for the rest of the main body text.

6) “2.1. Magnetic biocatalyst synthesis” (lines 76-96). It may be desirable to add a schematic representation to illustrate all the chemical steps followed by the authors. Eventually, it could be placed as Supplementary information (as the currently Scheme S1).

7) RESULTS & DISCUSSION. “3.1. Characterization of MNA” (lines 111-121). Did the authors estimate the mean diameter dimensions of the magnetic nanoparticle aggregates? How many magnetic NPs features were counted for this purpose? Same comment for the section “3.3 Characterization of MNA-CSP-3” (lines 159-200).

8) “We performed kinetic tests (…) followed by the Lineweaver-Burk graphs” (lines 206-208). Could the authors show the physical equation related to this type of analysis? This information could be placed as Supplementary Information (in the section concerning the “kinetic measurments”).

9) Figure 6 (line 237). Please, the standadr deviation (SD) bars should be added for each tested conditions. Then, some statistical analysis should be devoted to discern if the observed differences among these conditions are statistically different. Same comment for the Fig. S4 and Fig. S5 (Supplementary Information).

10) CONCLUSIONS. This section perfectly remarks the most relevant outcomes found by the authors in this field. The authors should add a brief statement to discuss about the future line actions to pursue this research and the open perspectives.

Comments on the Quality of English Language

The manuscript is generally well-written albeit it may be desirable if the authors could recheck it in order to polish some final details susceptible to be improved.

Author Response

Comment 1: ABSTRACT. “This results in (…)” (line 19). Please, the authors should modify this statement by “These results in (…)”. Even if the manuscript is generally well-written some revision is required in order to polish final details.

Response 1: The revised manuscript has been proofread by a native English speaker (and a chemist) so we believe the language has been polished.

Comment 2:  KEYWORDS. The authors should consider to add the term “magnetic nanoparticle aggregates” in the keyword list.

Response 2: The key words have been added.

 Comment 3:  INTRODUCTION. “When magnetic supports are based on magnetic nanoparticles (NPs) (…) amino groups” (lines 36-38). Here, even if I agree with the authors some further discussion needs to be devoted in order to highlight the importance of the shape of magnetic nanoparticles and how other alternative morphologies like cubic [1] or rod-shape [2] exhibit lower magnetic responses [3] than the shape chosen by the authors. This will significantly aid to the potential readers to better understand the significance of the magnetic nanoparticle synthesis strategy chosen by the authors.

Response 3: We would agree with the reviewer if we would deal with individual NPs when the shape is very important for magnetic properties. However, here we deal with large NP aggregates, so the morphology of individual NPs is rather a minor factor. The properties are determined by the ensemble of numerous NPs. Nevertheless, we added the suggested discussion and the references in the revised manuscript (page 1, lines 36-41).

Comment 4: “In this paper (…) magnetite (Fe3O4) NP aggregates via a precipitation route (…) magnetic NPs” (lines 60-64). What are the main advantages of magnetic NP aggregates in comparison to indiviual magnetic NP? A brief discussion should be furnished in this regard.

Response 4: It should be noted that the precipitation method always results in NP aggregates because no stabilizing media is added during the synthesis. The major advantage of aggregates is a higher magnetic response than that of individual NPs. This discussion is added on page 3 (lines 142-144) of the revised manuscript.

Comment 5: MATERIALS AND METHODS. “MNA (…) co-precipitation of FeCl2 and FeCl3 in (…)” (lines 77-79). Please, the authors should state the full-name of all chemical compounds the first time that they appear in the manuscript. This comment should be taken into account for the rest of the main body text.

Response 5: The full names of all chemicals have been provided throughout the manuscript.

Comment 6: “2.1. Magnetic biocatalyst synthesis” (lines 76-96). It may be desirable to add a schematic representation to illustrate all the chemical steps followed by the authors. Eventually, it could be placed as Supplementary information (as the currently Scheme S1).

Response 6: Please note that Scheme 1 fully depicts the magnetic biocatalyst synthesis except for the first step – MNA formation – which is trivial. Because this synthesis is well known and described, we believe the depiction of this step is not needed.

Comment 7: RESULTS & DISCUSSION. “3.1. Characterization of MNA” (lines 111-121). Did the authors estimate the mean diameter dimensions of the magnetic nanoparticle aggregates? How many magnetic NPs features were counted for this purpose? Same comment for the section “3.3 Characterization of MNA-CSP-3” (lines 159-200).

Response 7: We did estimate mean particle sizes using the grain-intercept method. For this, we used about 30 NPs for each sample. The aggregates are comparatively large (~ up to several microns) and polydisperse. The individual NP sizes vary between 5 nm and 30 nm with a mean diameter of about 12 nm. However, because all NPs are glued to each other these measurements are not accurate. This information has been added in the revised manuscript on page 3 (lines 144-149).

Comment 8: “We performed kinetic tests (…) followed by the Lineweaver-Burk graphs” (lines 206-208). Could the authors show the physical equation related to this type of analysis? This information could be placed as Supplementary Information (in the section concerning the “kinetic measurments”).

Response 8: The equation has been provided in the Supplementary Materials.

Comment 9: Figure 6 (line 237). Please, the standadr deviation (SD) bars should be added for each tested conditions. Then, some statistical analysis should be devoted to discern if the observed differences among these conditions are statistically different. Same comment for the Fig. S4 and Fig. S5 (Supplementary Information).

Response 9: The standard deviations have been added for all relevant figures.

Comment 10: CONCLUSIONS. This section perfectly remarks the most relevant outcomes found by the authors in this field. The authors should add a brief statement to discuss about the future line actions to pursue this research and the open perspectives.

Response 10: The conclusions have been extended to discuss outlook.

Round 2

Reviewer 1 Report

Comments and Suggestions for Authors

The manuscript has been revised. The introduction has been expanded and the new draft is more informative. The conclusion has been modified.

I agree with the corrections and think that the paper could be published in its present form.

Reviewer 3 Report

Comments and Suggestions for Authors

Authors have clarified the reviewer's doubts and acceptance is suggested. 

Reviewer 4 Report

Comments and Suggestions for Authors

The authors did a great effort in order to cover all the suggestions raised by the reviewers. For this reason, the scientific manuscript quality was greatly improved. Based on the significance of the most relevant outcomes found in this research and the scope of Nanomaterials, I warmly endorse this work for further publication in this journal.